# A concerted probiotic activity to inhibit periodontitis-associated bacteria

**Paul Mathias Jansen, Mohamed M. H. Abdelbary, Georg Conrads** *

Division of Oral Microbiology and Immunology, Department of Operative Dentistry, Periodontology and Preventive Dentistry, Rheinisch-Westfälische Technische Hochschule (RWTH) Aachen University Hospital, Aachen, Germany

* gconrads@ukaachen.de

**Data Availability Statement:** All relevant data are within the manuscript and its Supporting Information files.

**Funding:** The authors received no specific funding for this work.

## Abstract

Periodontitis can result in tooth loss and the associated chronic inflammation can provoke several severe systemic health risks. Adjunctive to mechanical treatment of periodontitis and as alternatives to antibiotics, the use of probiotic bacteria was suggested. In this study, the inhibitory effect of the probiotic *Streptococcus salivarius* subsp. *salivarius* strains M18 and K12, *Streptococcus oralis* subsp. *dentisani* 7746, and *Lactobacillus reuteri* ATCC PTA 5289 on anaerobic periodontal bacteria and *Aggregatibacter actinomycetemcomitans* was tested. Rarely included in other studies, we also quantified the inverse effect of pathogens on probiotic growth. Probiotics and periodontal pathogens were co-incubated anaerobically in a mixture of autoclaved saliva and brain heart infusion broth. The resulting genome numbers of the pathogens and of the probiotics were measured by quantitative real-time PCR. Mixtures of the streptococcal probiotics were also used to determine their synergistic, additive, or antagonistic effects. The overall best inhibitor of the periodontal pathogens was *L. reuteri* ATCC PTA 5289, but the effect is coenzyme B12-, anaerobiosis-, as well as glycerol-dependent, and further modulated by *L. reuteri* strain DSM 17938. Notably, in absence of glycerol, the pathogen-inhibitory effect could even turn into a growth spurt. Among the streptococci tested, *S. salivarius* M18 had the most constant inhibitory potential against all pathogens, followed by K12 and *S. dentisani* 7746, with the latter still having significant inhibitory effects on *P. intermedia* and *A. actinomycetemcomitans*. Overall, mixtures of the streptococcal probiotics did inhibit the growth of the pathogens equally or–in the case of *A. actinomycetemcomitans*- better than the individual strains. *P. gingivalis* and *F. nucleatum* were best inhibited by pure cultures of *S. salivarius* K12 or *S. salivarius* M18, respectively. Testing inverse effects, the growth of *S. salivarius* M18 was enhanced when incubated with the periodontal pathogens minus/plus other probiotics. In contrast, *S. oralis* subsp. *dentisani* 7746 was not much influenced by the pathogens. Instead, it was significantly inhibited by the presence of other streptococcal probiotics. In conclusion, despite some natural limits such as persistence, the full potential for probiotic treatment is by far not utilized yet. Especially, further exploring concerted activity by combining synergistic strains, together with the application of oral prebiotics and essential supplements and conditions, is mandatory.

**Competing interests:** The authors have declared that no competing interests exist.

## Introduction

Periodontal diseases are one of the most prevalent diseases in the world and can result in chronic inflammation and tooth loss [1]. Beside these oral symptoms, prolonged chronic to severe cases are associated with several systemic diseases such as atherosclerosis and diabetes mellitus [2]. Periodontitis, as the irreversible and most prevalent form of periodontal diseases, is caused by a dysbiotic shift in the subgingival biofilm frequently provoking an immune response. This interplay is the driving force for the chronic inflammation. Several bacterial species such as *Aggregatibacter actinomycetemcomitans*, *Fusobacterium nucleatum*, *Porphyromonas gingivalis*, *Prevotella intermedia*, and fastidious anaerobes including *Tannerella forsythia* and *Treponema denticola* are implicated in these shifts [3].

Probiotic bacteria have been suggested as an alternative or supplement to the conventional treatment of periodontitis, which includes mechanical removal of biofilm and antibiotics in severe (high stage, high grade) cases [4]. Due to the dramatic increase of antimicrobial resistance, many efforts are made to reduce prescription of antibiotics and to change from therapy to prevention. Probiotics might help to prevent infectious diseases or microbial dysbiosis, with periodontal diseases as a popular target because of its high prevalence and risk of sequelae (Global Burden of Disease Study, [5]). Two groups of probiotics were subject of this study, three streptococcal strains releasing cell membrane directed antimicrobial peptides called bacteriocins, and *Lactobacillus reuteri* producing 3-hydroxypropanal (reuterin).

*Streptococcus salivarius* subspecies *salivarius* (abbreviated henceforward as *S. salivarius*) strains K12 and M18 (Mia) were already explored as anti-pharyngitis, anti-caries and anti-halitosis probiotics [6–8]. Furthermore, K12 and M18 both significantly inhibited the expression of the cytokines IL-6 and IL-8 of gingival fibroblasts, which both induce inflammation when co-incubated with *P. gingivalis*, *F. nucleatum*, and *A. actinomycetemcomitans* [9]. Two strains each of *P. gingivalis* and *P. intermedia* were inhibited by K12 and M18 using the simultaneous antagonism method [10]. However, no inhibition of *P. gingivalis* and *P. intermedia* was observed when using a deferred antagonism test with *S. salivarius* M18 [11]. *S. salivarius* K12 showed antibacterial effects on *P. gingivalis*, *F. nucleatum*, and *A. actinomycetemcomitans* in a liquid co-culture, albeit the concentration of the strain K12 had to be higher than the concentration of the periodontal pathogens [12]. Concentrated supernatants of *Streptococcus oralis* subspecies *dentisani* (abbreviated henceforward as *S. dentisani*) strain 7746, a probiotic primarily explored to inhibit cariogenic mutans streptococci, inhibited the growth of *P. intermedia* and *F. nucleatum* and changes in the cell wall of both species were observed [13]. Esteban-Fernández et al. observed a growth reduction rate of 35 to 38% of *F. nucleatum* and *P. gingivalis* by supernatants of *S. dentisani* 7746. They observed cell lysis of *F. nucleatum* and the formation of vesicles in *P. gingivalis*. *S. dentisani* 7746 also inhibited the adherence of *F. nucleatum* and *P. gingivalis* to gingival cells. It decreased the production of cytokines after exposure to *F. nucleatum* and *P. gingivalis* and, therefore, potentially lowered the inflammation in gingival tissue [14]. On the other hand, Conrads et al. did not observe any significant inhibition of *P. gingivalis*, *P. intermedia* and *F. nucleatum* by *S. dentisani* 7746 and only a weak inhibition of *A. actinomycetemcomitans* applying agar diffusion tests [15]. These discrepancies were explained with differences in the inhibitory potential of different (adapted, mutated) subcultures or clones of *S. dentisani* 7746, but also due to individual test-conditions.

Finally, *Lactobacillus reuteri*, a common colonizer of the human gastrointestinal tract [16], is used as a probiotic in humans and animals [17–19]. The antimicrobial effects of *L. reuteri* are mainly, but not exclusively, based on a released substance termed reuterin (3-hydroxypropanal, 3-HPA, synonym 3(β) hydroxypropionaldehyde) with a broad-spectrum antimicrobial activity including gram-positive and gram-negative bacteria as well as yeasts, fungi, and even

protozoa [20]. It is an intermediate in the metabolism of glycerol to 1,3-propanediol catalyzed by the coenzyme B12-dependent diol dehydrase (oxygen-sensitive and membrane-associated glycerol dehydratase). If not released, 3-HPA is reduced by the (NADH)+H+-dependent-1,3-propanediol-oxydoreductase regenerating NAD+ [21–23]. The production is rate-limited because, if overproduced, reuterin is toxic for the producer cell [21]. Schaefer et al. proposed that the antimicrobial effects of reuterin are based on exerting oxidative stress and its interactions with thiol groups [24]. *L. reuteri* has already been used as an oral probiotic in the treatment of periodontitis [25] and a commercial product is marketed (Sunstar GUM®
PERIOBALANCE®, containing Prodentis® *L. reuteri* strains DSM 17938 and ATCC PTA 5289). In vivo studies were performed to assess the potential as a treatment for periodontal disease. Vivekananda et al. showed a significant reduction of several periodontitis indicators, such as periodontal pocket depth (PPD), gingival index (GI), plaque index (PI), gingival bleeding index (GBI), and clinical attachment level (CAL), after adjunct treatment with *L. reuteri*. The levels of *P. intermedia*, *P. gingivalis*, and *A. actinomycetemcomitans* were also significantly reduced after the treatment [26]. A study that determined the effects on gingivitis was done by Iniesta et al. with *L. reuteri* tablets [27]. The authors found no significant reduction of GI & PI compared to the placebo group but, however, they observed a significant reduction of *P. intermedia* in saliva and *P. gingivalis* in subgingival samples. In another study, Teughels et al. observed a reduction in pocket depth compared to mechanical treatment (scaling and root planning, SRP) alone. Also, lower numbers of *P. gingivalis* were detected after treatment with *L. reuteri* lozenges than with the placebo [25].

The aim of this study was to investigate single and combinatory inhibitory effects of streptococcal strains K12, M18, and 7746, as well as *L. reuteri* strains ATCC PTA 5289 and DSM 17938 on bacterial species associated with periodontal disease. The probiotics and pathogens were co-incubated in a simple model mimicking the conditions in a periodontal pocket. The resulting cell numbers of the periodontal pathogens and probiotics were determined by a quantitative real-time PCR (qRT-PCR). Genome numbers were analyzed to estimate the inhibitory effects of a specific probiotic strain or combination of probiotic strains on a particular periodontal pathogen. Also and rarely tested, the reverse effect of the pathogens on the growth of the probiotics was investigated. Our null hypothesis was that combinations of strains do not significantly increase the probiotic effect ($\alpha$ = 5%).

## Materials and methods

### Bacterial cultures

*S. salivarius* subsp. *salivarius* K12, *Lactobacillus reuteri* ATCC PTA 5289, and *L. reuteri* DSM 17938 were isolated from lozenges of the corresponding marketed products, namely BLIS K12 Throat Guard® (BLIS Technologies Limited, Wellington, New Zealand) and PERIOBA-LANCE® (Sunstar Europe SA, Switzerland). Based on GenBank genome sequence data available (GU564004.1 in case of ATCC PTA 5289, CP002844.1 in case of DSM 17938, the latter a plasmid-free progeny of strain SD2112) identity of both *L. reuteri* strains was confirmed by a multidrug ABC transporter gene directed PCR. *Streptococcus salivarius* subsp. *salivarius* M18 was kindly provided by R. Lütticken (Aachen, Germany) and *S. oralis* subsp. *dentisani* 7746 by A. Mira (Valencia, Spain). All strains (probiotic producers and periodontal test strains from our own collection) were grown on tryptic soy agar with sheep blood (TSASB, Oxoid Germany) for 48 hours at 37˚C. Different atmospheric conditions were chosen. All probiotic strains were initially incubated in an atmosphere with 7–8% $CO_2$. The periodontopathogenic strains *Porphyromonas gingivalis* ATCC 33277, *Prevotella intermedia* ATCC 25611, and *Fusobacterium nucleatum* ATCC 25586 were grown anaerobically in a GasPak™ EZ anaerobe

pouch system, whereas *Aggregatibacter actinomycetemcomitans* ATCC 33384 was cultivated in a candle jar (MART Anaerobic jar). Stock suspensions of the probiotic strains with known concentrations of colony-forming units (cfu) were prepared and deep shock frozen (-73°C) in a mixture of 0.9% NaCl (850 μl) and glycerol (150 μl; Merck). The same aliquot used throughout all experiments. Fresh liquid cultures of the pathogenic strains were used to ensure optimal fitness. Colonies from the blood agar plates were used to inoculate 5 ml of a brain-heart-infusion (BHI) broth containing 50 μl of a vitamin K-hemin solution (Becton Dickinson) and incubated either anaerobically or microaerophilic (candle jar in case of *A. actinomycetemcomitans*) at 37°C for 48 hours.

## Growth inhibition of the periodontal pathogens by probiotic strains and vice versa

A standard inhibition assay was developed to assess the inhibitory potential of probiotic strains against the periodontal pathogens. This was achieved by comparing the cell numbers (genome equivalents to be exact) of the pathogens after incubation with or without probiotic strains for 48 hours at 37°C anaerobically (in the case of anaerobes) or in a candle jar (in the case of *A. actinomycetemcomitans*). A 1:1 mixture of twofold concentrated BHI broth and autoclaved human saliva (donated and pooled 1:1 from two healthy probands) was used as standard growth medium (SGM) for the inhibition assay. Each well of a 96 well cell culture plate (Greiner Bio-One) contained 100 μl of the medium. To assess the effect of glycerol on inhibition, *L. reuteri* strains were cultivated with (1% w/v) or without glycerol. The stock suspensions of the probiotic strains (see above; except *L. reuteri* DSM 17938) were used to inoculate the microtiter wells generally reaching a start concentration of $10^4$ cfu per 100 μl and well. Next, pathogenic strains were inoculated by adding 10 μl of the liquid pre-culture. The exact concentration of each inoculum was measured by quantitative real-time PCR. All tests were done in triplicates including the negative control and growth controls (without inhibiting probiotic strains). In order to find the most inhibiting probiotic formula, periodontal pathogens were co-incubated either with a singular probiotic strain or a mixture of two streptococcal or lactobacilli probiotics. However, mixtures of streptococci and lactobacilli were not tested so far to avoid genus-genus interactions and to ease interpretation. The overall concentration of all probiotic cells in a mixture was set to the maximum of $10^5$ cfu per ml. After completing the inhibition assay, the cells were harvested by transferring culture from wells to 1.5 ml Eppendorf microliter tubes. Scraping the biofilm fraction was supported by the pipette tip, which was slightly cut back producing a sharper edge, and by pipetting up and down for dispersion. After centrifugation, the supernatant was discarded and the pellet was resuspended in 100 μl 0.9% NaCl. For washing, the steps centrifugation, discarding of the supernatant, and resuspension of pellet were repeated. Samples were then stored at -73°C until further use. After thawing, the samples were centrifuged and the supernatant discarded. The resulting pellet was treated with 20 μl of a lysozyme & mutanolysin solution (LM, 15 mg lysozyme and 500U mutanolysin in 1ml TE-buffer) for 30 minutes at 37°C lysing bacterial cell walls. DNA was extracted with the QIAmp DNA Mini Kit according to the manufacturer instructions and samples were stored at -20°C.

A qRT-PCR was performed to measure the genome number of both, the pathogenic and the therapeutic probiotic strains, before (inoculum, t = 0) and after (t = 48h) inhibition assays. The DNA of the stock suspensions was serially diluted in tenfold step with nuclease-free water to create a standard curve. The qRT-PCR was performed by the aid of a QuantStudio 3 and in 96 well plate block formats (Thermo Fisher Scientific, Dreieich, Germany). Except for the reciprocal inhibition experiments (limited to duplicates), every pathogen/probiotic

**Table 1. List of primers used in this study.**

| Primer | Sequence 5'-3' | Amplicon Size | Ta [°C] | Reference |
|---|---|---|---|---|
| PF1-F | AGAGTTTGATCCTGGCTCAG | | 54–60 | [28] |
| Aa-R (combined with PF1) | GGCATGCTATTAACACAC | 469 bp | 54 | This study |
| Fn-R (combined with PF1) | GTCATCGTGCACACAGAATTGCTG | 360 bp | 60 | [29] |
| Pi-R (combined with PF1) | GTTGCGTGCACTCAAGTCCGCC | 660 bp | 56 | [30] |
| Pg-R (combined with PF1) | CAATACTCGTATCGCCCGTTATTC | 478 bp | 59 | [31] |
| SDent-16S-F | TGAAGGAGGAGCTTGCTTCTC | | 59 | [15] |
| SDent-16S-R | CAAACAGTTATCATGCAATAACTG | 137 bp | 59 | [15] |
| Ssal-M18-C4-F2 | GAGGTCCGGTTAATGGTTGT | | 54 | This study |
| Ssal-M18-C4-R | CTATGCTGGAGATGACGG | 252 bp | 54 | This study |

Ta = annealing temperature of primer.

combination and controls were run in biological triplicates and the DNA extracted from each well (template) was measured in technical triplicates. The PowerUp™ SYBR™ Green Master Mix (Thermo Fisher Scientific) was used to create a reaction mix. Each well contained 20 μl of the reaction mix with the following components: PowerUp™ SYBR™ Green Master Mix (10 μl), Forward Primer (0.1 μl), Reverse Primer (0.1 μl), nuclease-free water (8.8 μl), Template (1 μl). The concentrations of all primers (synthesized by TIB Molbiol, Berlin, Germany) were 100 μM. DNA of all pathogens and–for inverse experiments—two probiotics (M18, 7746) was amplified and quantified with strain specific primers (Table 1). As a negative control, nuclease-free water was added instead of the template.

The qRT-PCR was performed with the following temperature profile: initial denaturation at 95°C (2 min); 40 cycles of: 95°C (15 s), $T_a$ (see Table 1, 15 sec), 72°C (60 s); and final elongation at 72°C (10 min).

### Effect of glycerol on periodontal pathogens and on *Lactobacillus reuteri*

Since glycerol was added as substrate to *L. reuteri* cultures for reuterin (3-hydroxypropanal) production the sole influence of glycerol on the pathogen growth had to be investigated. The periodontal pathogens were grown in SGM with 1% w/v or without glycerol. The rest of the experiment was performed as described in the previous section and the cell density was measured by qRT-PCR.

*L. reuteri* ATCC PTA 5289 in combination with the *L. reuteri* strain DSM 17938 in commercially available lozenges (Sunstar GUM® PERIOBALANCE®) had been used for the treatment of periodontitis by re-establishing a healthy biofilm and combating dysbiosis [25]. Thus, the synergistic effects of these two lactobacilli strains on *F. nucleatum* were exemplarily investigated. *F. nucleatum* was chosen as its genome numbers varied the most in testing's with or without glycerol. The SGM with 1% w/v or without glycerol was inoculated with $10^4$ cfu of *F. nucleatum* ATCC 25586. Except for the control, the different wells were inoculated with *L. reuteri* ATCC PTA 5289, *L. reuteri* DSM 17938 or a 1:1 mixture of both strains reaching a cell density of each probiotic of $10^5$ cfu/ml in a volume of 100 μl and the plates were incubated at 37°C in an anaerobic atmosphere for 48 hours. Anaerobiosis is important, as not only *F. nucleatum*, but also the reuterin-producing enzyme glycerol dehydratase, is oxygen-sensitive. The

DNA of cultures was isolated and the cell density of *F. nucleatum* again measured by qRT-PCR.

## Statistical analysis

Data (bacterial cell or genome counts, respectively) were analyzed using GraphPad Prism (version 8.4.3; San Diego, CA). All data were not normally (not Gaussian) distributed. With non-pairing data, the non-parametric one-way ANOVA (Kruskal-Wallis) test was performed. Generally, the mean rank of columns (culture result) was compared to the mean rank of a control column (culture without probiotic). As the magnitude of genome counts varied over several logs, the uncorrected Dunn's test was preferred. Correction of multiple comparisons was, however, also performed and differences were reported.

## Results

### Growth inhibition of the periodontal pathogens by probiotic strains

Without the addition of probiotics, the periodontal pathogens reached genome numbers between $1.14^*10^6$ (*F. nucleatum*) and $1.74^*10^8$ (*P. gingivalis*) after 48h of incubation (Fig 1A–1D). In principle, all probiotic strains inhibited the growth of periodontal pathogens, except *L. reuteri* in case that no glycerol was added to the medium. However, no probiotic strain or strain-combination–so far tested—was able to reduce the pathogen much below $10^5$ genomes. Glycerol for itself did only slightly, non-significantly, inhibit the growth of test strains (S1 Fig). The individual impact is reported below, firstly for single strains and secondly for strain combinations. The statistical results presented are based on an uncorrected Dunn's Kruskall Wallis test, thus each comparison stands alone. Correction for multiple comparisons generally doubled the p-value and in case of p < .05 significance was lost.

The *P. intermedia* ATCC 25611 growth control reached an average genome number of $1.66^*10^6$, calculated from three measurements of three independent biological replicates (Fig 1A). The co-incubation with the streptococcal probiotics *S. salivarius* M18, *S. salivarius* K12, or *S. dentisani* 7746 resulted in a lowered growth of *P. intermedia* (M18 = $4.87^*10^5$ genomes; K12 = $2.20^*10^5$ genomes; 7746 = $1.41^*10^5$ genomes), with the latter reaching significance (p < .05). The inhibitory potential of *L. reuteri* ATCC PTA 5289 was dependent on the presence of glycerol in the growth medium. The overall strongest inhibition of *P. intermedia* was observed in co-incubation with strain PTA 5289 when glycerol was added to the medium ($2.02^*10^4$ genomes, p < .001). In contrast, in the absence of glycerol, *P. intermedia* was not inhibited at all by PTA 5289 reaching $1.63^*10^6$ genomes per well. The effect of glycerol supplementation was highly significant (p < .001).

The mean *P. gingivalis* ATCC 33277 genome number was $1.74^*10^8$ in the growth control (Fig 1B). Both *S. salivarius* strains, M18 and K12, did inhibit the growth of *P. gingivalis* but only K12 significantly (M18 = $8.43^*10^7$ genomes; K12 = $4.73^*10^7$ genomes, p < .05). In contrast *S. dentisani* 7746 had no inhibitory effect on *P. gingivalis*, still reaching $1.06^*10^8$ genomes. Compared to the control, the genome numbers of *P. gingivalis* were even higher when grown in the presence of *L. reuteri* ATCC PTA 5289 without glycerol ($2.48^*10^8$ genomes). However, the addition of glycerol resulted in a significant reduction of *P. gingivalis* if co-incubated with the PTA 5289 strain ($1.55^*10^7$ genomes, p < .01). Again, the effect of glycerol supplementation was highly significant (p < .001).

The mean *F. nucleatum* ATCC 25586 genome number in the growth control was $1.14^*10^6$ after 48h (Fig 1C). The co-incubation with *S. dentisani* 7746 resulted in a mean cell count of $8.49^*10^5$ genomes, slightly below the control. *S. salivarius* M18 inhibited the growth of *F. nucleatum* ($3.72^*10^5$ genomes), but not reaching significance. The co-incubation with *S. salivarius* K12 had no inhibitory effect ($1.19^*10^6$ genomes). Co-incubation with *L. reuteri* ATCC

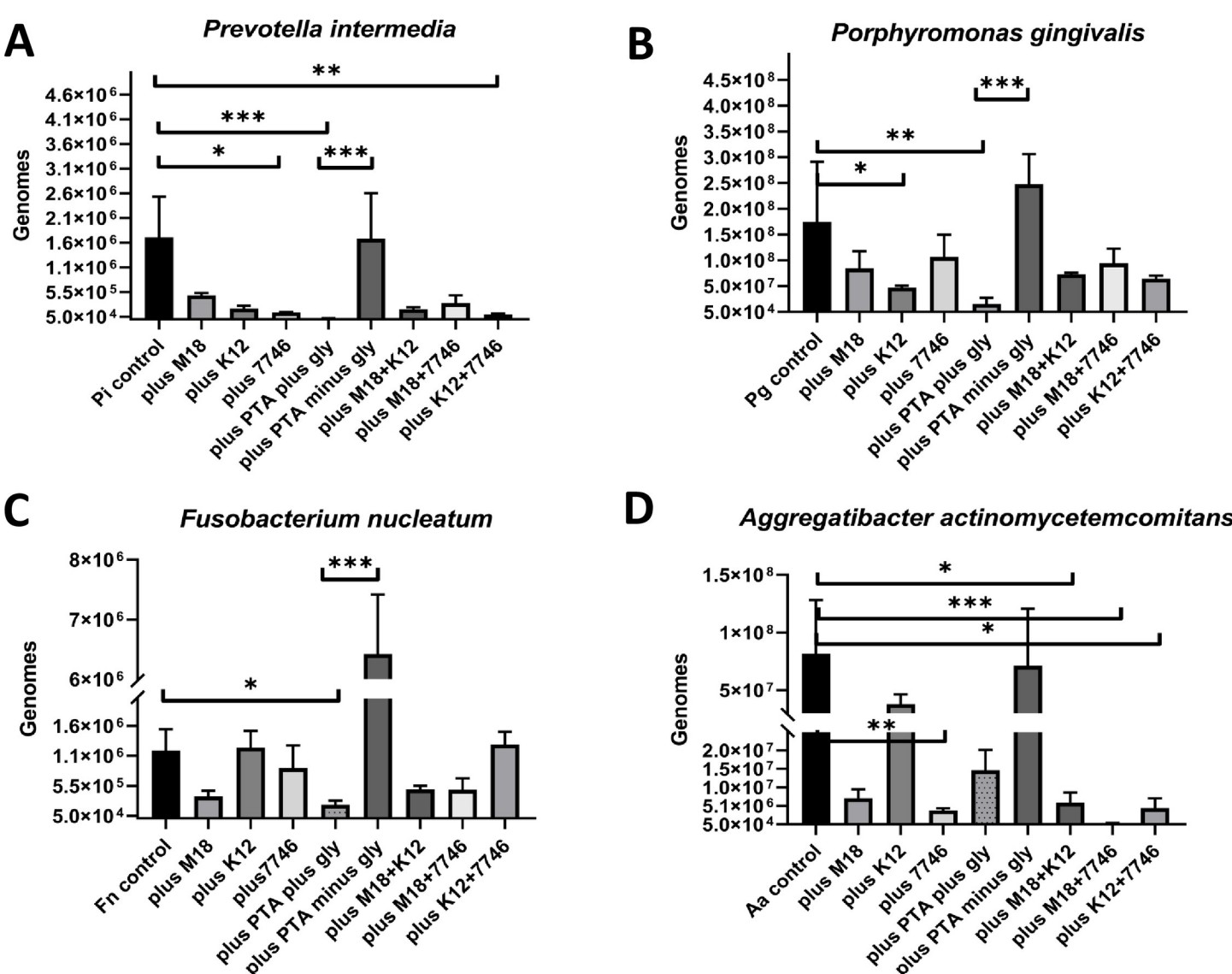

**Fig 1. Inhibition experiments demonstrating the *in vitro* probiotic potential of *Streptococcus salivarius* M18 and K12, *S. dentisani* 7746, and *Lactobacillus reuteri* ATCC PTA 5289 (plus/minus glycerol) on the growth of four (A-D) different periodontal pathogens.** All experiments were performed in biological triplicates and the DNA measured in technical triplicates. Statistical significance was calculated based on Kruskal-Wallis test. Level of significance *p < .05, **p < .01, ***p < .001. Abbreviations: Pi (*Prevotella intermedia*), Pg (*Porphyromonas gingivalis*), Fn (*Fusobacterium nucleatum*), Aa (*Aggregatibacter actinomycetemcomitans*), M18 (*S. salivarius* subsp. *salivarius* M18), K12 (*S. salivarius* subsp. *salivarius* K12), PTA (*Lactobacillus reuteri* ATCC PTA 5289), 7746 (*Streptococcus oralis* subsp. *dentisani* 7746), gly (glycerol).

PTA 5289 in a medium with glycerol reduced *F. nucleatum* significantly ($2.29^*10^5$ genomes, p < .05) but without glycerol supplementation the pathogen genome number increased more than 5-fold ($6.42^*10^6$ genomes, p < .001).

Finally, the mean number of *A. actinomycetemcomitans* ATCC 33384, measured after 48 hours of microaerophilic incubation in the growth control, was $8.17^*10^7$ genomes (Fig 1D). The highest inhibition by streptococci was observed when co-incubated with *S. dentisani* 7746 (drop to $3.82^*10^6$ genomes, p < .01). Here and in contrast to the obligate anaerobic test strains, *L. reuteri* ATCC PTA 5289 resulted in a non-significant reduction of *A. actinomycetemcomitans*, even after glycerol supplementation, possibly because of the oxygen-sensitive nature of the glycerol dehydratase.

Next, the effect of mixtures of the oral streptococcal probiotics *S. salivarius* M18, *S. salivarius* K12, and *S. dentisani* 7746 on the growth of the periodontal pathogens was investigated to assess synergistic, additive, or dilutive (antagonistic) inhibitory effects compared to addition of single probiotic strains (Fig 1A–1D, last three columns).

A strong inhibition of *P. intermedia* was observed when co-incubated with a mixture of *S. salivarius* K12 and *S. dentisani* 7746 ($9.90^*10^4$ genomes, p < .01). In comparison to the inhibitory effects of the two single probiotics ($2.20^*10^5$ in case of K12, p = .08 and $1.41^*10^5$ in the case of 7746, p < .05), the significance was higher indicating a synergistic effect. A mixture of both *S. salivarius* strains (K12 and M18) or of *S. salivarius* M18 and *S. dentisani* 7746 also inhibited the growth of *P. intermedia* but with no significant additive or synergistic effect.

All mixtures of streptococcal probiotics lowered the genome numbers of *P. gingivalis* compared to the growth control after 48h (Fig 1B). However, mixing the best streptococcal probiotic, namely K12, with either M18 or 7746, showed an antagonistic effect. As a result, no mixture reached the inhibitory and significant power of K12 applied as a single probiotic.

Only two of the three mixtures inhibited the growth of *F. nucleatum* (Fig 1C). The best inhibitor was the mixture of both *S. salivarius* strains M18 and K12 reducing *F. nucleatum* down to $4.91^*10^5$ genomes, followed by the mixture of M18 and *S. dentisani* 7746 reducing to $4.83^*10^5$ genomes. Both co-inhibitors, however, diluted the probiotic effect of sole M18 which was $3.72^*10^5$ genomes for comparison. Interestingly, the mixture of K12 & 7746 slightly enhanced the growth of *F. nucleatum* in comparison to the control ($1.24^*10^6$ genomes versus $1.14^*10^6$ genomes), but this growth stimulation was not significant.

Finally, every mixture of the streptococcal probiotics inhibited the growth of *A. actinomycetemcomitans* significantly (Fig 1D). The overall lowest number was observed when co-incubated with a mixture of *S. salivarius* M18 & *S. dentisani* 7746 ($3.01^*10^5$ genomes), indicating a significant synergistic effect (p < .001). The increase of inhibition between applying individual versus mixed probiotics was also significant (p < .01). The mixture of both *S. salivarius* strains, in an additive manner, significantly (p < .05) reduced *A. actinomycetemcomitans* down to a mean of $5.90^*10^6$ genomes. Worth mentioning, M18 compensated or even over-compensated the somewhat weak inhibitory effect of K12 here. In contrast, the mixture K12 / 7746 resulted in $4.44^*10^6$ *A. actinomycetemcomitans* genomes, indicating that the 7746-inhibitory effect was slightly diluted by K12.

All inhibitions were also calculated as percentages (Table 2) and graphically visualized in S2 Fig. Clearly, *L. reuteri* ATCC PTA 5289 plus glycerol, as a single probiotic strain, had the best anti-pathogen effect. However, with depletion of glycerol, as essential for reuterin production, the inhibition can turn into a growth spurt (negative inhibition marked in bold in Table 2 and under the 0-line in S2 Fig).

**Table 2. Inhibition (in %) of four different periodontopathogenic species by four oral probiotic strains and single or in different combinations.**

| | Ss M18 | Ss K12 | Sd 7746 | Lr PTA plus gly | Lr PTA minus gly | M18 & K12 | M18 & 7746 | K12 & 7746 |
|---|---|---|---|---|---|---|---|---|
| *Prevotella intermedia* | 70.7 | 87.4 | 91.7 | 98.8 | 4.0 | 88.0 | 81.3 | 93.9 |
| *Porphyromonas gingivalis* | 55.2 | 73.3 | 36.3 | 92.1 | **-33.6** | 60.5 | 48.5 | 65.6 |
| *Fusobacterium nucleatum* | 64.1 | -9.04 | 22.8 | 79.6 | **-533** | 57.6 | 57.2 | -20.1 |
| *Aggregatibacter actinomycetemcomitans* | 91.7 | 59.9 | 95.1 | 80.7 | 27.8 | 93.7 | 99.4 | 94.9 |
| **Mean** | 70.44 | 52.90 | 61.47 | 87.82 | **-133.9** | 74.94 | 71.61 | 58.55 |
| **SD**[*] | 13.47 | 37.06 | 32.32 | 8.01 | 231.84 | 16.05 | 20.06 | 46.89 |

[*]As the results of four different species are integrated the standard deviation (SD) is naturally high. Abbreviations: M18 (*S. salivarius* subsp. *salivarius* M18), K12 (*S. salivarius* subsp. *salivarius* K12), PTA (*Lactobacillus reuteri* ATCC PTA 5289), 7746 (*Streptococcus oralis* subsp. *dentisani* 7746), gly (glycerol).

### Growth inhibition of streptococcal probiotics by periodontal pathogens

Since the interaction between probiotic bacteria and periodontal pathogens is not one-sided, the effects of pathogens on the growth of *S. salivarius* M18 & *S. dentisani* 7746 were exemplarily assessed again by qRT-PCR. The resulting streptococcal genome numbers were compared to the probiotic growth control without pathogen (Fig 2A and 2B).

The co-incubation with every periodontopathogenic strain used in this study enhanced the growth of *S. salivarius* M18 (Fig 2A), but only significantly in the case of *P. gingivalis* (p < .05). In contrast, without any gram-negative target organism, the mean genome number was $3.56^*10^5$. If co-incubated with *P. gingivalis* it reached $1.78^*10^6$ genomes or with *P. intermedia* it reached $1.54^*10^6$ genomes. The lowest growth-stimulation was detected when incubated with *A. actinomycetemcomitans* ($9.05^*10^5$ genomes).

The addition of another streptococcal probiotic (either *S. salivarius* K12 or *S. dentisani* 7746) to *S. salivarius* M18 plus any periodontal pathogen tested, generally (in 7 out of 8 cases) resulted in a further growth spurt of *S. salivarius* M18. The exception here was the mixture of *S. salivarius* M18 & K12 in the case of *P. intermedia*, which resulted in a lower genome number (decrease from $1.54^*10^6$ to $5.19^*10^5$ genomes) compared to the reference culture.

The genome numbers of *S. dentisani* 7746 were again higher compared to the growth control ($6.25^*10^6$) when co-incubated with either *P. gingivalis* ($1.40^*10^7$) or *A. actinomycetemcomitans* ($8.34^*10^6$) (Fig 2B). The co-incubation with *F. nucleatum* or *P. intermedia* slightly lowered the genome number of *S. dentisani* 7746 ($4.86^*10^6$ and $4.54^*10^6$). Overall, the periodontal pathogens used in this study had little, non-significant effects on the growth of *S.*

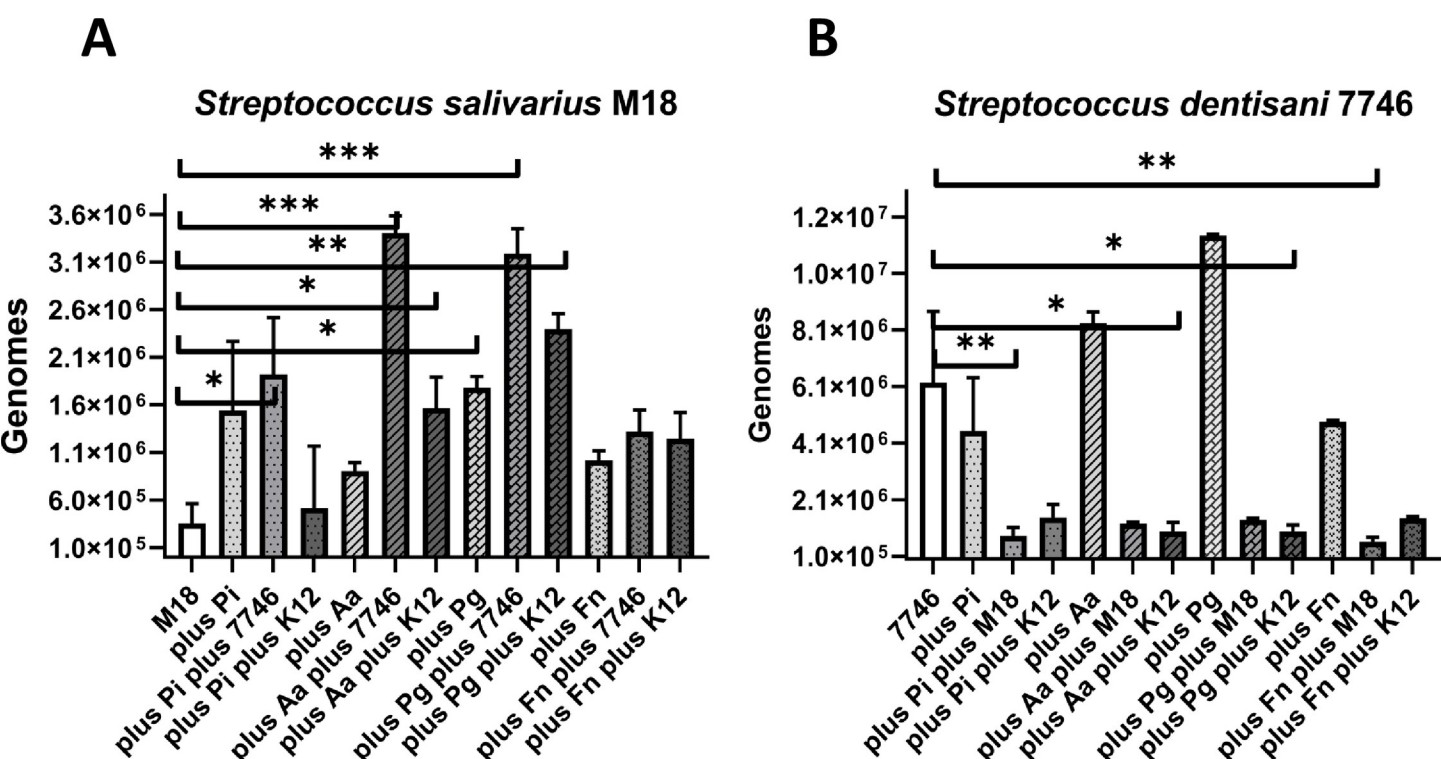

**Fig 2. Reciprocal inhibition experiments demonstrating the growth-stimulating effect of four periodontopathogenic bacteria (*Prevotella intermedia* Pi, *Aggregatibacter actinomycetemcomitans* Aa, *Porphyromonas gingivalis* Pg, *Fusobacterium nucleatum* Fn) on *Streptococcus salivarius* M18 (A) and *S. dentisani* 7746 (B).** All experiments were performed in biological duplicates and the DNA measured in technical triplicates. Level of significance *p < .05, **p < .01, ***p < .001. Abbreviations: Pi (*Prevotella intermedia*), Pg (*Porphyromonas gingivalis*), Fn (*Fusobacterium nucleatum*), Aa (*Aggregatibacter actinomycetemcomitans*), K12 (*S. salivarius* subsp. *salivarius* K12).

*dentisani* 7746. With other words, the gram-negative targets did neither inhibit nor stimulate the growth of this probiotic. Interestingly, the overall genome numbers of 7746 were significantly (p < .05-.01) or by trend (p≤.17) lowered when additionally co-incubated with either *S. salivarius* M18 or K12, probably demonstrating a risk for antagonistic effects of other streptococcal strains.

### Effect of glycerol on periodontal pathogens and on *Lactobacillus reuteri*

As outlined above, glycerol had an essential influence on the inhibitory effect of *L. reuteri* ATCC PTA 5289. For one, the inhibition of the three obligate anaerobic periodontal pathogens was highly significantly (p < .001) stronger when the medium was supplemented with 1% w/v glycerol. As an adverse effect, the growth of *P. gingivalis* (by 33.6%) and *F. nucleatum* (by 533%) was even stimulated when incubated with strain PTA 5289 in a glycerol-free culture (Table 2). Therefore, the sole effects of glycerol on the growth of periodontal pathogens had to be identified (S1 Fig). In principle, *F. nucleatum*, *P. intermedia*, *P. gingivalis* and *A. actinomycetemcomitans* did grow without ($1.49^*10^6$; $1.14^*10^7$; $1.89^*10^8$; and $7.44^*10^6$ respectively) or with ($1.25^*10^6$; $1.45^*10^6$; $1.04^*10^8$; and $6.08^*10^5$ respectively) 1% w/v glycerol. Although the numbers of periodontal pathogens were reduced by glycerol, the difference did not reach significance (S1 Fig). Taken all results together, the chemical effect of the essential glycerol has to be subtracted from the probiotic effect of *Lactobacillus reuteri* in glycerol-rich cultures.

Since strain *L. reuteri* ATCC PTA 5289 was isolated from lozenges (Sunstar GUM® PERIOBALANCE®), that also contained another strain, *L. reuteri* DSM 17938, the synergistic effect of both strains was examined, again with and without the addition of glycerol (Fig 3). These tests were exemplarily performed with *F. nucleatum* ATCC 25586, as it showed the most pronounced influence of glycerol. Every well was inoculated with $6.2^*10^5$ *F. nucleatum* cells. After 48 hours, the mean genome-numbers of the growth control without glycerol ($6.05^*10^6$) were higher than with glycerol ($4.90^*10^6$), which was correspondent to the results of the aforementioned experiment. The co-incubation with strain PTA 5289 did significantly (p < .05) lower the number of *F. nucleatum* down to $9.16^*10^5$ genomes when the medium contained glycerol. However, after subtracting the sole inhibitory effect of glycerol, significance was just missed (p = .0578). Without glycerol, strain PTA 5289 enhanced *F. nucleatum* growth reaching $8.60^*10^6$ genomes. A reason for the upregulation in absence of glycerol could be catabolism gene-upregulation in *F. nucleatum* after coaggregation, as recently found in a co-culture with *Streptococcus gordonii* [32]. Interestingly, the sole incubation with strain DSM 17938 reduced *F. nucleatum* with and without glycerol ($1.53^*10^6$ and $1.78^*10^6$ genomes, respectively), pointing on an inhibitory mechanism independent of the glycerol-depending reuterin-effect. Finally, a combination of both *L. reuteri* strains ATCC PTA 5289 and DSM 17938 inhibited *F. nucleatum*, reducing the effect of glycerol to a non-significant level ($3.69^*10^6$ versus $1.04^*10^6$ genomes, with p = 0.1722). Inhibition of the probiotic strain mixture, however, reached the same magnitude as strain PTA 5289 alone. Thus, strain DSM 17938 seems to act as a helper strain being especially supportive in case of glycerol-shortage or absence.

### Discussion

With a rising skepticism to use antibiotics in the treatment of chronic inflammatory diseases, such as periodontal diseases, alternatives including phage-therapy or probiotic approaches are gaining more attention. Several probiotics to re-establish a healthy, eubiotic oral microbiome have been isolated and explored. Among them are *Firmicutes*, including streptococci and lactobacilli, the most promising candidates. Recently, we have published about the *in vitro* anti-cariogenic potential of *Streptococcus oralis* subspecies *dentisani* 7746 in comparison to other

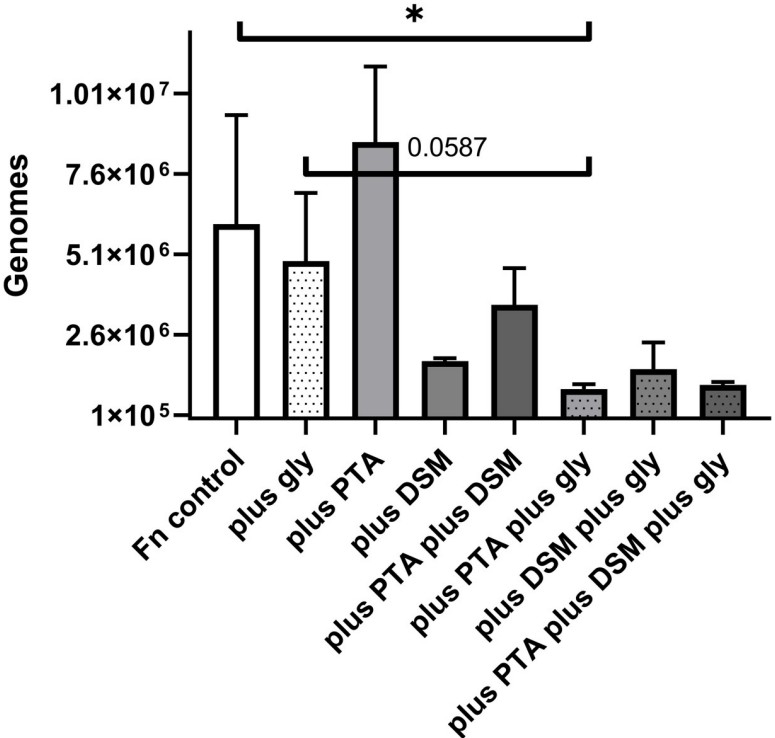

**Fig 3. Inhibition experiments demonstrating the *in vitro* probiotic potential of *Lactobacillus reuteri* ATCC PTA 5289, DSM 17938, and a combination of both (minus/plus glycerol) on the growth of *F. nucleatum* ATCC 25586.** Level of significance *p < .05. Abbreviations: Fn (*Fusobacterium nucleatum*), PTA (*Lactobacillus reuteri* ATCC PTA 5289), DSM (*Lactobacillus reuteri* DSM 17938), gly (glycerol).

probiotic strains or combinations of strains [15]. Here, we explored the *in vitro* anti-periodontitis potential of this species in comparison to *Streptococcus salivarius* subsp. *salivarius* M18 and K12, as well as *Lactobacillus reuteri* strains ATCC PTA 5289 and DSM 17938. The inhibitory effect was tested on four of the most prominent and cultivable periodontal pathogens, namely *Aggregatibacter actinomycetemcomitans*, *Fusobacterium nucleatum*, *Prevotella intermedia*, and the key pathogen *Porphyromonas gingivalis*. As isolated from different research groups, so far their probiotic activity was tested only individually and applying different antagonist-tests, making the direct comparison of probiotic power difficult. Here for the first time, strains 7746, M18, K12 (separately and in combination), and PTA 5289 (plus/minus DSM 17938) were tested in parallel and under exactly the same conditions (BHI: saliva 1: 1 medium, same atmospheric conditions, duration of 48h). Some combinations, such as mixtures of streptococcal and lactobacilli strains or mixtures of more than two strains, were excluded, keeping the culture conditions more controlled and easing the interpretation. In future experiments, more interactions based on the results here, should be tested.

Regarding the mostly bacteriocin-driven inhibitory potential of streptococcal strain mixtures in comparison to individual strains, it can be concluded from our results that additive or dilutive (antagonistic) inhibitory effects are more likely to occur than synergistic effects. Our null hypothesis, that combinations of strains do not significantly increase their probiotic effects, was approved. However, there were at least two exceptions found. A strong inhibitory

effect of *P. intermedia* was observed when co-incubated with a mixture of *S. salivarius* K12 and *S. dentisani* 7746. In comparison to the inhibitory effects of the two single probiotics, the significance was higher, indicating a synergistic effect. Furthermore, every mixture of the streptococcal probiotics improved *A. actinomycetemcomitans* growth-inhibition significantly. The overall lowest number was observed when co-incubated with a mixture of *S. salivarius* M18 and *S. dentisani* 7746, again indicating a concerted, synergistic effect. The increase of inhibition between applying individual versus mixed probiotics was also significant (p < .01). In contrast, for *P. gingivalis*, no mixture reached the inhibitory and significant power of K12 and for *F. nucleatum* both co-inhibitors diluted the probiotic effect of sole M18. In conclusion here, a concerted probiotic effect of otherwise individually explored streptococcal strains is possible, at least to target some pathogens. It is known that *S. dentisani* 7746 produces about ten different bacteriocins [15]. Combined with the plasmid-encoded salivaricin-variants A2, B, 9, and MPS produced by K12 and M18 [33], as well as bacteriocin immunity factors, a concerted activity directed against the pathogens is needed avoiding a friendly fire of against the co-producer.

As the interaction pathogen-probiotic is bidirectional, we tested the influence of pathogens on the growth of *S. salivarius* M18 & *S. dentisani* 7746 exemplarily. The co-incubation with every periodontopathogenic strain stimulated the growth of *S. salivarius* M18, significantly in the case of *P. gingivalis*. This was observed for the first time here. A possible explanation could be that a dying and disintegrating target provides more nutrients and supplements for the producer. In contrast, the co-incubation with pathogens had only non-significant effects on the growth of *S. dentisani* 7746.

Regarding the inhibitory potential *of L. reuteri* strains or strain mixtures on *F. nucleatum*, clearly pure *L. reuteri* ATCC PTA 5289 had the best anti-pathogen effect. By way of exception, *A. actinomycetemcomitans* was more reuterin-resistant and M18-susceptible, which had been reported before for culture supernatants of these probiotics [34]. Critically, with depletion of the essential glycerol the inhibition can turn into a growth spurt. The combination with *L. reuteri* DSM 17938, a putative immunomodulatory strain [35, 36], seems also to reduce the glycerol-dependency, an additional synergistic effect for the Sunstar GUM® PERIOBALANCE® Prodentis® mixture, overlooked so far. There is much to discuss about the reuterin (3-HPA) production of these strains. Many of the essential parameters are not taken into account when testing or applying *L. reuteri*, especially in clinical studies. It must be emphasized that 3-HPA production by the enzyme glycerol dehydratase is at least dependent on i) presence of substrate glycerol, ii) anaerobiosis as the enzyme is oxygen-sensitive, and iii) presence of traces of B12 [21–24, 37]. If one of these three conditions is not fulfilled, the probiotic activity will remain weak. On the other hand, if all conditions are perfectly concerted, the pathogen inhibition can be enhanced. In our experiments, we fulfilled these conditions in all wells (measurements) by adding 1% glycerol (versus control) and most plates were incubated anaerobically in a Brain-Heart-Infusion-broth with natural saliva containing B12-traces. However, the reduced reuterin-activity after microaerophilic incubation with *A. actinomycetemcomitans* could already be an indicator for glycerol dehydratase inhibition by traces of oxygen. Fortunately, B12 is heat-stable and the activity is not reduced much by autoclavation [38]. However, the exact B12-concentration was not determined in our experiment and we did not test whether addition of B12 would have further stimulated the reuterin-effect.

Underrating these conditions could be one reason why clinical studies on the anti-periodontitis activity revealed very different results, supporting [25, 39–43], intermediate [44, 45], or not-supporting [46–48] its application, and recently leading to rejection of a health claim [10]. This could further implicate that reuterin production needs a deep, anaerobic pocket and might thus not be helpful in mild cases. However, as many highly oxygen-sensitive, obligate

anaerobic pathogens are even present in shallowed periodontal pockets or in the healthy sulcus, it seems that—wherever anaerobic pathogens are able to multiply—reuterin will be produced as oxygen-concentration is locally low. As reported in a process engineering study, for the efficient reuterin production, the presence of a certain glycerol concentration is critical, as enough substance should be produced but too much antimicrobial reuterin is toxic for the producer [22]. Translated for clinical studies and applications, an optimal glycerol concentration should be ensured during probiotic therapy, at least initially or–after the strain is established in the mouth—in phases of inflammation. To our knowledge this condition was never considered enough in any clinical study. The probiotic lozenges were given as produced and marketed. And the same might be true for B12-supplementation, a condition easily to fulfill as this vitamin is cheap and stable. So far, PERIOBALANCE® does contain–besides bacteria and peppermint flavor—an isomalt cryoprotectant, emulsifier, sweetener, and a few percentages of hydrogenated palm oil. As least palm oil is a source of triacylglycerole which can be degraded by bacteria releasing glycerol. Other glycerol-sources are blood and tissue in the inflamed gingiva. However, it might be helpful to measure glycerol in situ and–if suboptimal for reuterin production–to further supplement during probiotic therapy, ideally combined with B12.

Besides the direct antimicrobial activity subjected in our study, there are three additional central questions to answer when applying (concerted) probiotics.

Firstly, could lactic acid bacteria be cariogenic? The probiotics discussed here are acidogenic and aciduric and often found to be associated with caries. And while it can be effective in clearing periodontal bacteria in vitro or even in vivo, other impacts on the overall oral ecology and caries-etiology are important to consider [15, 49].

Secondly, is a reduction by 2 or 3 log-steps, usually measured by *in vitro* studies (including the present) enough to keep a chronic inflammatory reaction under control *in vivo*? This can only be answered in non-commercially biased, randomized, double blinded, placebo-controlled, and prognostic clinical studies with many subjects. At least for the application of *L. reuteri* such studies do exist but show, as outlined above, ambivalent results. A reason for hope here is, comparable to experience with antibiotics, that a reduction by only a few log-steps still gives the immune system the chance to take over, turning a vicious circle into an upwards spiral. The addition of immunostimulating strains might further improve the probiotic concert.

Thirdly, how long will the probiotic strain or a combination persist *in situ* and thus act at a particular oral side? For *S. salivarius* K12, in a former study, we found persistence on different mucosae for as long as three weeks, but with steadily decreasing numbers after day eight [7]. Other studies addressing this issue are very rare, as usually only the abundance before and shortly after end of treatment—but not the persistence—is measured. For instance, persistence of *S. dentisani* 7746 at mucosal membranes was never addressed before. In a study by Burton et al. the oral cavity "persistence" of *S. salivarius* M18 was investigated in 75 subjects receiving four different doses for 28 days [10]. The last testing for M18 was done as usual after the last administration, challenging the measurability of true persistence. The authors conclude that the percentage of subjects having the M18 strain detected in their saliva first increased with the dose, but after day seven slowly dropped down [10]. For gut flora, the clearance of probiotics is better studied. In rats it was shown that, from five different probiotic strains (*Lactobacillus acidophilus* LA742, *Lactobacillus rhamnosus* L2H, *Bifidobacterium lactis* HN019 and the oral probiotic *S. salivarius* K12) *B. lactis* and *L. rhamnosus* persisted seven days, but the oral K12 was already non-detectable at day three [50]. The latter result could be due to the "wrong" intestinal niche. Vice versa, intestinal *L. reuteri* might have reduced persistence in the oral cavity. Romani Vestman et al. conducted a randomized, double-blinded, placebo-controlled study with a 6-week PERIOBALANCE® intervention period and 3- and 6-month follow-up, investigating the effects on regrowth of mutans streptococci after full-mouth disinfection. *L. reuteri*

was frequently detected by culture during the intervention period but in only three test group subjects (10%) at follow-ups and in low numbers [51]. Indeed, the persistence of intestinal *L. reuteri* in the oral niche seems to be weak. As a matter of fact, clearance of niche-foreign bacteria is a basic defense principle in natural microbial ecosystems and this is why we call such intruders "transient". A concerted probiotic activity should therefore, besides pathogen inhibition, also consider immunostimulation for pathogen clearance (e.g. by addition of *L. reuteri* DSM 17938) and niche-persistence (e.g. by addition of true residential strains linking others).

In conclusion, investigating the beneficial effects of probiotic bacteria to maintain or re-establish oral health is a very important topic. The full potential for probiotic treatment is by far not utilized yet. Especially, further exploring the concerted force on three levels, pathogen-inhibition, immunostimulation, and niche-persistence, together with the application of oral prebiotics and essential supplements and conditions, is desirable.

## Supporting information

**S1 Fig. Influence of 1% (w/v) glycerol on the growth of four periodontopathogenic bacteria.** Abbreviations: Pi (*Prevotella intermedia*), Pg (*Porphyromonas gingivalis*), Fn (*Fusobacterium nucleatum*), Aa (*Aggregatibacter actinomycetemcomitans*), gly (glycerol).
(TIF)

**S2 Fig. Overall *in vitro* percent inhibition of periodontal pathogens by probiotic strains *S. salivarius* M18 and K12, *S. dentisani* 7746, and *L. reuteri* ATCC PTA 5289.** In a culture without glycerol, *L. reuteri* caused a growth spurt (negative inhibition) of some pathogens. For data see Table 2. Abbreviations: M18 (*S. salivarius* subsp. *salivarius* M18), K12 (*S. salivarius* subsp. *salivarius* K12), PTA (*Lactobacillus reuteri* ATCC PTA 5289), 7746 (*Streptococcus oralis* subsp. *dentisani* 7746), gly (glycerol).
(TIF)

## Acknowledgments

We would like to thank Mrs. Beate Melzer-Krick for her excellent technical assistant and Rudolf Lütticken for providing strain M18 and editing the manuscript.

## Author Contributions

**Conceptualization:** Georg Conrads.

**Data curation:** Mohamed M. H. Abdelbary.

**Investigation:** Paul Mathias Jansen.

**Methodology:** Paul Mathias Jansen, Mohamed M. H. Abdelbary.

**Supervision:** Georg Conrads.

**Validation:** Georg Conrads.

**Writing – original draft:** Paul Mathias Jansen.

**Writing – review & editing:** Mohamed M. H. Abdelbary, Georg Conrads.

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
