## [Decision Letter · Decision Letter 0]

5 Feb 2021

PONE-D-20-35741

A concerted probiotic activity to inhibit periodontitis-associated bacteria

PLOS ONE

Dear Dr. Georg Conrads,

Thank you for submitting your manuscript to PLOS ONE. After careful consideration, we feel that it has merit but does not fully meet PLOS ONE’s publication criteria as it currently stands. Therefore, we invite you to submit a revised version of the manuscript that addresses the points raised during the review process.

We look forward to receiving your revised manuscript.

Kind regards,

Abdelwahab Omri, Pharm B, Ph.D

Academic Editor

PLOS ONE

Journal Requirements:

Reviewers' comments:

Reviewer's Responses to Questions

**Comments to the Author**

1. Is the manuscript technically sound, and do the data support the conclusions?

Reviewer #1: Yes

Reviewer #2: Yes

Reviewer #3: Yes

2. Has the statistical analysis been performed appropriately and rigorously? 

Reviewer #1: Yes

Reviewer #2: Yes

Reviewer #3: Yes

3. Have the authors made all data underlying the findings in their manuscript fully available?

Reviewer #1: Yes

Reviewer #2: Yes

Reviewer #3: Yes

4. Is the manuscript presented in an intelligible fashion and written in standard English?

Reviewer #1: Yes

Reviewer #2: Yes

Reviewer #3: No

5. Review Comments to the Author

Reviewer #1: This is an interesting paper that addresses an important topic, the inhibition of periodontal pathogens by commensals and probiotics. While the methodology is fairly straightforward, the novelty of work lies in the testing of combinations of organisms and examining the impact of the pathogenic organisms on the commensals. The dependence of L. reuteri on glycerol for an antagonistic effect is also an important advance.

Specific comments

1. Discussion of non-statistically significant results should be limited and more cautious. For example, its not clear that the inhibitory potential of K12 was diluted by M18 (line 262), when none of the conditions were statistically significant.

2. There is no need to repeat numbers in the text which are presented in the figures and table.

3. Line 112-113. The authors should state a biological hypothesis rather than a statistical one.

4. Introduction. the meaning of 'a common abbreviation used' is unclear.

Reviewer #2: Overview: The manuscript “A concerted probiotic activity to inhibit periodontitis-associated bacteria” describes a straightforward approach to characterize effects of several probiotic bacteria on periodontal pathogenic bacteria in vitro. The authors performed co-cultures of strains, with single and double strains of probiotics cultured with each pathogen, then analyzed growth of each strain after 48 h by measuring abundance of genomic DNA via qRT-PCR. In general, this was an interesting study that certainly has some applications in dental hygiene. I have a few minor comments and one major comment about the manuscript and research.

Minor comments:

1. The manuscript is well written, and English is very good. However, a careful review by a native English speaker would help to polish the language.

2. Lines 112-113: “Our null hypothesis was that there is no significant difference of pure culture versus co-culture.” For clarity, what difference are you referring to? It would also be nice if you could come back to this hypothesis at the end of the paper and state whether or not it was confirmed.

3. Lines 248-249: “without glycerol supplementation the pathogen cell number increased more than 5-fold…” Why would this happen? Can you provide some more details as to why the pathogens would grow higher in the PTA – glycerol condition than on their own?

4. Table 2: The order of the Lr PTA w/o gly and Lr PTA with gly is swapped from Fig. 1. That was a bit confusing to me for a second. I recommend swapping the two columns to be consistent with the order in Fig. 1.

5. Figure 2- I’m not sure if this is necessary. It may be better placed in the Supplementary Info, as it is mostly a repeat of the data in Table 2.

6. Lines 320-321: “Thus, perfect for the application as a probiotic…” This sentence is rather confusing and could benefit from being re-worded to improve clarity.

Major issues:

1. Lines 117-119. The authors isolated three different strains of bacteria from commercially available lozenges. However, there is no description of how or whether the isolates were characterized and confirmed. Were the isolates verified to be the strains they report? If so, please explain how the validation performed. If not, the strains should be verified.

Reviewer #3: This study assessed the probiotic effects of S. salivarius strains M18 and K12, S. oralis subsp. dentisani 7746, or L. reuteri ATCC PTA 5289 (or a combination thereof) on periodontal pathogens, P. gingivalis ATCC33277, F. nucleatum ATCC 25586, P. intermedia ATCC 25611 and A. actinomycetemcomitans ATCC 33384. L. reuteri glycerol-dependent antagonistic properties (production of reuterin) was also examined. The study also investigated the impact of the periopathogens on the growth of the probiotic strains. All analysis was done via qRT-PCR. Overall, the paper is straight forward and presents relatively new information. Experiments performed also appear sound and appropriately controlled. There are several aspects of the paper that could be improved though. The resolution of the figures should be improved; they appear to be of poor quality. Figure 2 does not present any new data so should be omitted. The discussion section also lacks some important and relevant information. While it states that L. reuteri had the “best anti-pathogen effect” and discusses in detail what aspects could be improved as a probiotic therapy, it is also worth mentioning that Lactobacilli can be acidogenic and aciduric in certain conditions and are often found to be associated with caries. And while it can be effective in clearing periodontal bacteria in vitro, the impact it can have to the overall oral ecology is important to note. Additionally, the discussion section specifically could be improved if several grammar issues can be corrected with further proofreading/editing (some examples are listed below).

1. Line 139 and throughout the paper: growth inhibition was assessed quantifying bacterial genomes via qPCR. Hence, phrases like“comparing cell numbers” or referring to the detected bacterial genome as cell numbers is inaccurate (since CFUs weren’t done to actually count cell numbers).

2. Line 140: were biofilms grown under anaerobic conditions? If so, please specify here. It is mentioned in the discussion section/line 433 but not specified here.

3. Line 141: The growth condition authors used/BHI both and autoclaved human saliva. Please include a reference or explain why this condition was selected.

4. Line 154: Collection of biofilms. The author states cells were simply “harvested by transferring culture from wells”. Was any scraping of the biofilm with subsequent sonication step involved?

5. It could help readers follow the results better if figures in Figure 1 are labeled A-D, and referenced in text/results section

6. Line 259: Authors state P. intermedia inhibition is enhanced via a synergistic effect of K12+7746, compared to single species. Is there a significant decrease of P.intermedius genome in K12+7746 vs. K12 or 7746 only?

7. Line 259 and on: Instead of organizing the results section this way, it would help if all the findings of each graph within Figure 1 is written together (instead of splitting single probiotic species impact on specific periodontal bacteria and then going back later to stating the results on mixed-species impact on that periodontal bacteria)

8. Figure 2 presents nothing new. Table 2 is fine the way it is but Figure 2 does not really add to the manuscript. Figure 2 should be removed.

9. Line 321: “producer-activation” ? Are authors referring more detection of genome as “activation”? Sentence should be rewritten. Referring it to as “persistence” of that probiotic strain seems more accurate.

10. Line 344-345: Please add “respectively” after the numbers in the parentheses

11. Line 345: “However” should be changed to “Although” or “While” and omit “but”

12. Figure 4: Title of figure should be F. nucleatum, as the format in Figure 1, since the bacterial genome reported are of F. nucleatum when grown with two different strains of L. reuteri.

13. Line 358: remove “Anyway” and also throughout the manuscript

14. Line 375: please change “gain more attention” to “are gaining more attention”

15. Line 391: Omit however

16. Line 411-413: Just simply state it as a possible explanation (take out “an explanation for this,…cannot be given”).

17. Line 435: Should be “the activity is not reduced much by autoclaving”

18. Line 435-436: Change to “..was not determined in our experiment and we did not test whether addition of B12 would have further..”

19. Line 445: “Learning from biotechnology”…? As reported in the literature…?

6. PLOS authors have the option to publish the peer review history of their article (what does this mean?). If published, this will include your full peer review and any attached files.

Reviewer #1: No

Reviewer #2: No

Reviewer #3: No

---

## [Author Response · Author response to Decision Letter 0]

22 Feb 2021

Dear Editor, 

Thank you for your letter dated February 5th. 2021 summarizing the reviewer’s comments. We have carefully reviewed the comments and have revised the manuscript accordingly. Our responses are given in a point-by-point manner below.

We have updated the PLOS ONE's style requirements.

We deleted the paragraph referring to “data not shown”: From “Because of a slight cross-reactivity between K12- (data therefore not shown)…” until …the streptococcal probiotics tested reduce the periodontal key-pathogen P. gingivalis and may even benefit from its killing.”

We have checked the statistic again.

We hope the revised version is now suitable for publication and are looking forward to hearing from you.

Reviewer #1: This is an interesting paper that addresses an important topic, the inhibition of periodontal pathogens by commensals and probiotics. While the methodology is fairly straightforward, the novelty of work lies in the testing of combinations of organisms and examining the impact of the pathogenic organisms on the commensals. The dependence of L. reuteri on glycerol for an antagonistic effect is also an important advance.

Authors: Thank you very much for your supportive words.

Specific comments

1. Discussion of non-statistically significant results should be limited and more cautious. For example, its not clear that the inhibitory potential of K12 was diluted by M18 (line 262), when none of the conditions were statistically significant.

Authors: You are right that this might be an over-interpretation. We changed the text accordingly, now reading:

Lines 262ff: A mixture of both S. salivarius strains (K12 and M18) or of S. salivarius M18 and S. dentisani 7746 also inhibited the growth of P. intermedia, but with no significant additive or synergistic effect.

2. There is no need to repeat numbers in the text which are presented in the figures and table.

Authors: We disagree regarding numbers in figures. The exact count is hard to deduce from a figure but the exact number is still important for direct comparison in the text. Regarding tables, we now avoid repetition, except for a few very important data. Thank you.

3. Line 112-113. The authors should state a biological hypothesis rather than a statistical one.

Authors: We like this deep-minded comment and our suggestion is shown below and integrated at the end of introduction.

Lines 112-113: Our null hypothesis was that combinations of strains do not significantly increase the probiotic effect (� = 5%).

4. Introduction. the meaning of 'a common abbreviation used' is unclear.

Authors: We agree and have found an alternative way to show the correct taxonomy without constantly using the very long, bulky (subspecies level) names.

Line 60: Streptococcus salivarius subspecies salivarius (abbreviated henceforward as S. salivarius) strains K12…

Line 69: Streptococcus oralis subspecies dentisani (abbreviated henceforward as S. dentisani) strain 7746…

Reviewer #2: Overview: The manuscript “A concerted probiotic activity to inhibit periodontitis-associated bacteria” describes a straightforward approach to characterize effects of several probiotic bacteria on periodontal pathogenic bacteria in vitro. The authors performed co-cultures of strains, with single and double strains of probiotics cultured with each pathogen, then analyzed growth of each strain after 48 h by measuring abundance of genomic DNA via qRT-PCR. In general, this was an interesting study that certainly has some applications in dental hygiene. I have a few minor comments and one major comment about the manuscript and research.

Authors: Thank you very much for your supportive words.

Minor comments:

1. The manuscript is well written, and English is very good. However, a careful review by a native English speaker would help to polish the language.

Authors: We have polished the language and Reviewer #3 helped to improve the grammar. 

2. Lines 112-113: “Our null hypothesis was that there is no significant difference of pure culture versus co-culture.” For clarity, what difference are you referring to? It would also be nice if you could come back to this hypothesis at the end of the paper and state whether or not it was confirmed.

Authors: Thank you for this suggestion. We have modified the 0-hypothesis and have added a sentence to the discussion:

Introduction: Lines 112-113: Our null hypothesis was that combinations of strains do not significantly increase the probiotic effect (� = 5%).

Discussion: Lines 391ff: Our null hypothesis, that combinations of strains do not significantly increase the probiotic effect, was approved. However, there were at least two exceptions found…

3. Lines 248-249: “without glycerol supplementation the pathogen cell number increased more than 5-fold…” Why would this happen? Can you provide some more details as to why the pathogens would grow higher in the PTA – glycerol condition than on their own?

Authors: This is a reasonable and central question. Knowledge about growth conditions and metabolism of Fusobacterium nucleatum might help to answer this question. Searching for this, we came through an article of Rogers et al. (Marsh-group) from 1991 (Oral Microbiol Immunol. 1991Aug;6(4):250-5.doi: 10.1111/j.1399-302x.1991.tb00486.x.). Firstly, we considered F. nucleatum as acidophilic and thus probably stimulated by lactic acid bacteria. Indeed, its pH-range is between pH 7.7 and as low as 5.8. But as pH 7.4 was found to be the optimum, acid tolerance cannot be the answer. Next, we searched for effects by coaggregation. F. nucleatum is considered as a ‘bridge‐organism’ that facilitates colonization of other bacteria by co-aggregation‐mediated mechanisms. Sharma et al. 2004 (https://doi.org/10.1111/j.1399-302X.2004.00175.x), investigating Tannerella forsythia-F. nucleatum co-cultures, stated that “synergistic biofilm formation between the two species was dependent on cell–cell contact and soluble components of the bacteria were not required”. Later, more information came from a study of Mutha et al. 2018 (Mol Oral Micro). They found that in a S. gordonii-F. nucleatum co-culture -by comparison with monocultures- 16 genes were (up-)regulated in F. nucleatum and even 119 genes were regulated in S. gordonii. In both species, genes involved in amino acid and carbohydrate metabolism were strongly affected by coaggregation. In particular, one particular operon in F. nucleatum, encoding sialic acid uptake and catabolism, was up-regulated 2- to 5-fold following coaggregation. We do not yet have a proof in our L. reuteri-F. nucleatum co-culture but a comparable up-regulation is our hypothesis.

We added the following text plus the Mutha et al 2018 reference: A reason for the upregulation in absence of glycerol could be catabolism gene-upregulation in F. nucleatum after coaggregation, as recently found in a co-culture with Streptococcus gordonii. 

4. Table 2: The order of the Lr PTA w/o gly and Lr PTA with gly is swapped from Fig. 1. That was a bit confusing to me for a second. I recommend swapping the two columns to be consistent with the order in Fig. 1.

Authors: We have edited Table 2. We also optimized consistency of order and naming throughout the text, Table and Figures.

5. Figure 2- I’m not sure if this is necessary. It may be better placed in the Supplementary Info, as it is mostly a repeat of the data in Table 2.

Authors: Figure 2 is important as presenting relative numbers and for all four pathogens. But data are indeed repeated in Table 2. As a compromise, we kept it as Supplementary S2 Figure.

6. Lines 320-321: “Thus, perfect for the application as a probiotic…” This sentence is rather confusing and could benefit from being re-worded to improve clarity.

Authors: We decided to delete this sentence. It was criticized by another reviewer too and it contains interpretation not appropriate as Results. Thank you.

Major issues:

1. Lines 117-119. The authors isolated three different strains of bacteria from commercially available lozenges. However, there is no description of how or whether the isolates were characterized and confirmed. Were the isolates verified to be the strains they report? If so, please explain how the validation performed. If not, the strains should be verified.

Authors: A very good question. We invested some time to ensure the identity of strains isolated from the products (a good source otherwise, because this is exactly what the consumers take). We see our efforts reflected in this question that we can answer confidentially: This task was easy in the case of K12 as this is the only strain in product BLIS K12. Furthermore, we successfully performed bacteriocin-gene-directed-PCR-experiments with this strain before (see Horz et al. 2007). However, because of two L. reuteri strains this task was more sophisticated in the Sunstar product. Fortunately, the genome sequences of both strains were available. We do not want to expand the M&M section too much, but we added one sentence for clarification: 

Lines 120ff: Based on GenBank genome sequence data available (GU564004.1 in case of ATCC PTA 5289, CP002844.1 in case of DSM 17938, the latter a plasmid-free progeny of strain SD2112) identity of both L. reuteri strains was confirmed by a multidrug ABC transporter gene directed PCR. 

We can provide primer data if needed.

Reviewer #3: This study assessed the probiotic effects of S. salivarius strains M18 and K12, S. oralis subsp. dentisani 7746, or L. reuteri ATCC PTA 5289 (or a combination thereof) on periodontal pathogens, P. gingivalis ATCC33277, F. nucleatum ATCC 25586, P. intermedia ATCC 25611 and A. actinomycetemcomitans ATCC 33384. L. reuteri glycerol-dependent antagonistic properties (production of reuterin) was also examined. The study also investigated the impact of the periopathogens on the growth of the probiotic strains. All analysis was done via qRT-PCR. Overall, the paper is straight forward and presents relatively new information. Experiments performed also appear sound and appropriately controlled. 

Authors: Thank you very much for your supportive words.

There are several aspects of the paper that could be improved though. 

A) The resolution of the figures should be improved; they appear to be of poor quality. 

Authors: Figures were directly exported from GraphPad Prism but using low-resolution preferences unfortunately. In the new version we provide .tif and .emf formats with the highest resolution. 

B) Figure 2 does not present any new data so should be omitted. 

Authors: In Figure 2 results of all four pathogens are integrated and presented as relative (%) inhibition. But there is redundancy with Table 2 indeed. We decided to keep it as Supplementary File (S2_Fig).

C) The discussion section also lacks some important and relevant information. While it states that L. reuteri had the “best anti-pathogen effect” and discusses in detail what aspects could be improved as a probiotic therapy, it is also worth mentioning that Lactobacilli can be acidogenic and aciduric in certain conditions and are often found to be associated with caries. And while it can be effective in clearing periodontal bacteria in vitro, the impact it can have to the overall oral ecology is important to note. 

Authors: We are thankful to the reviewer. In a former probiotic-article of our group (Conrads et al 2019) we actually discussed this janus face of acidogenic and aciduric probiotics in depths. We are citing this article (again) plus Badet 2008, the latter reviewing the ecological role of lactobacilli in the oral cavity. 

Furthermore, we are expanding the Discussion-section adding the following sentences: Besides the direct antimicrobial activity subjected in our study, there are three [was formerly two] additional central questions to answer when applying (concerted) probiotics. Firstly, could lactic acid bacteria be cariogenic? The probiotics discussed here are acidogenic and aciduric and often found to be associated with caries. And while it can be effective in clearing periodontal bacteria in vitro or even in vivo, other impacts on the overall oral ecology and caries-etiology are important to consider [15, 49].

D) Additionally, the discussion section specifically could be improved if several grammar issues can be corrected with further proofreading/editing (some examples are listed below).

Authors: Thanks for your time and engagement. It is welcome.

1. Line 139 and throughout the paper: growth inhibition was assessed quantifying bacterial genomes via qPCR. Hence, phrases like “comparing cell numbers” or referring to the detected bacterial genome as cell numbers is inaccurate (since CFUs weren’t done to actually count cell numbers).

Authors: You are right. However, qPCR standard curves are produced from DNA extracted from cultures of known cell (CFU) counts. What we count are thus genome-equivalents of cell numbers. For further clearness we used “genome(s)” in almost all cases and included the following phrase in line 141ff: This was achieved by comparing the cell numbers (genome equivalents to be exact) of the pathogens after incubation with or without probiotic strains for 48 hours at 37°C.

2. Line 140: were biofilms grown under anaerobic conditions? If so, please specify here. It is mentioned in the discussion section/line 433 but not specified here.

Authors: Thank you. We have specified: Line 141: …after incubation with or without probiotic strains for 48 hours at 37°C anaerobically (in the case of anaerobes) or in a candle jar (in the case of A. actinomycetemcomitans).

3. Line 141: The growth condition authors used/BHI both and autoclaved human saliva. Please include a reference or explain why this condition was selected.

Authors: The assay is new as stated in the text but based on our experience with autoclaved saliva as important adjuvant in a secondary caries model based on Streptococcus mutans (see Hetrodt et al. 2018, https://doi.org/10.1016/j.archoralbio.2018.03.013). Brain-Heart Infusion medium is a well-known rich medium to support growth of fastidious organisms.

4. Line 154: Collection of biofilms. The author states cells were simply “harvested by transferring culture from wells”. Was any scraping of the biofilm with subsequent sonication step involved?

Authors: We agree that these little tricks are actually very important and should be reported. We added a sentence explaining the procedure: Scraping the biofilm fraction was supported by the pipette tip, which was slightly cut back producing a sharper edge, and by pipetting up and down for dispersion.

5. It could help readers follow the results better if figures in Figure 1 are labeled A-D, and referenced in text/results section

Authors: Thank you, we have changed accordingly! To be congruent, we used the format for Fig 2 (A-B) as well.

6. Line 259: Authors state P. intermedia inhibition is enhanced via a synergistic effect of K12+7746, compared to single species. Is there a significant decrease of P.intermedius genome in K12+7746 vs. K12 or 7746 only?

Authors: Yes, the significance changed from ns (K12 only, p=0.08) and p<0.05 (7746 only) to p<0.01 (both strains), but we re-wrote this sentence for clarification: A strong inhibition of P. intermedia was observed when co-incubated with a mixture of S. salivarius K12 and S. dentisani 7746 (9.90*104 genomes, p<.01). In comparison to the inhibitory effects of the two single probiotics (2.20*105 in case of K12, p=0.08 and 1.41*105 in the case of 7746, p<.05), the significance was higher indicating a synergistic effect.

7. Line 259 and on: Instead of organizing the results section this way, it would help if all the findings of each graph within Figure 1 is written together (instead of splitting single probiotic species impact on specific periodontal bacteria and then going back later to stating the results on mixed-species impact on that periodontal bacteria)

Authors: We think that this is a matter of taste. As the other reviewers agreed with this order, we would like to keep it sorted single versus mixture activity. 

8. Figure 2 presents nothing new. Table 2 is fine the way it is but Figure 2 does not really add to the manuscript. Figure 2 should be removed.

Authors: As this was noted by another reviewer we decided to reformat it as Supplementary Files (S2_ Fig).

9. Line 321: “producer-activation” ? Are authors referring more detection of genome as “activation”? Sentence should be rewritten. Referring it to as “persistence” of that probiotic strain seems more accurate.

Authors: We decided to delete this sentence. It also contains interpretation not appropriate under “Results”.

10. Line 344-345: Please add “respectively” after the numbers in the parentheses

Authors: Thank you, we have changed accordingly (line 342 now).

11. Line 345: “However” should be changed to “Although” or “While” and omit “but”

Authors: Thank you, we have changed accordingly.

12. Figure 4: Title of figure should be F. nucleatum, as the format in Figure 1, since the bacterial genome reported are of F. nucleatum when grown with two different strains of L. reuteri.

Authors: Thank you, a good point actually. We have changed accordingly.

13. Line 358: remove “Anyway” and also throughout the manuscript

Authors: Thank you. We have found and removed twice.

14. Line 375: please change “gain more attention” to “are gaining more attention”

Authors: We have changed the grammar.

15. Line 391: Omit however

Authors: We have omitted “however”.

16. Line 411-413: Just simply state it as a possible explanation (take out “an explanation for this,…cannot be given”).

Authors: Thank you for editing this important sentence. We have changed into “A possible explanation could be that a dying and disintegrating target provides more nutrients and supplements for the producer.”

17. Line 435: Should be “the activity is not reduced much by autoclaving”

Authors: Thank you for polishing the grammar; we acknowledge this.

18. Line 435-436: Change to “..was not determined in our experiment and we did not test whether addition of B12 would have further..”

Authors: Thank you again for polishing the grammar; we still acknowledge this.

19. Line 445: “Learning from biotechnology”…? As reported in the literature…?

Authors: We want to express that medical microbiology can learn from other fields such as process engineering. We changed the phrase into “As reported in a process engineering study,..”

---

## [Editor Report · Decision Letter 1]

24 Feb 2021

A concerted probiotic activity to inhibit periodontitis-associated bacteria

PONE-D-20-35741R1

Dear Dr. Georg Conrads,

We’re pleased to inform you that your manuscript has been judged scientifically suitable for publication and will be formally accepted for publication once it meets all outstanding technical requirements.

Kind regards,

Abdelwahab Omri, Pharm B, Ph.D

Academic Editor

PLOS ONE

---

## [Editor Report · Acceptance letter]

26 Feb 2021

PONE-D-20-35741R1 

A concerted probiotic activity to inhibit periodontitis-associated bacteria 

Dear Dr. Conrads:

I'm pleased to inform you that your manuscript has been deemed suitable for publication in PLOS ONE. Congratulations! Your manuscript is now with our production department. 

Kind regards, 

on behalf of

Dr. Abdelwahab Omri 

Academic Editor

PLOS ONE